# Environmental Cadmium Exposure and Dental Indices in Orthodontic Patients

**DOI:** 10.3390/healthcare9040413

**Published:** 2021-04-02

**Authors:** Hui-Ling Chen, Jason Chen-Chieh Fang, Chia-Jung Chang, Ti-Feng Wu, I-Kuan Wang, Jen-Fen Fu, Ya-Ching Huang, Ju-Shao Yen, Cheng-Hao Weng, Tzung-Hai Yen

**Affiliations:** 1Department of Dentistry and Craniofacial Orthodontics, Chang Gung Memorial Hospital, Linkou 333, Taiwan; ma3608@cgmh.org.tw (H.-L.C.); s610523@gmail.com (C.-J.C.); patrick1992815@gmail.com (T.-F.W.); 2School of Medicine, College of Medicine, Chung Shan Medical University, Taichung 402, Taiwan; jasonfang100@gmail.com; 3Department of Nephrology, China Medical University Hospital, Taichung 404, Taiwan; ikwang@mail.cmuh.org.tw; 4College of Medicine, China Medical University, Taichung 406, Taiwan; 5Department of Medical Research, Chang Gung Memorial Hospital, Linkou 333, Taiwan; cgfujf@cgmh.org.tw; 6Graduate Institute of Clinical Medical Sciences, Chang Gung University, Taoyuan 333, Taiwan; 7Department of Laboratory Medicine, Chang Gung Memorial Hospital, Linkou 333, Taiwan; hycymm@cgmh.org.tw; 8Department of Medical Biotechnology and Laboratory Science, College of Medicine, Chang Gung University, Taoyuan 333, Taiwan; 9Department of Nephrology, Chang Gung Memorial Hospital, Linkou 333, Taiwan; q311@cgmh.org.tw (J.-S.Y.); drweng@seed.net.tw (C.-H.W.); 10Clinical Poison Center, Kidney Research Center, Center for Tissue Engineering, Chang Gung Memorial Hospital, Linkou 333, Taiwan; 11College of Medicine, Chang Gung University, Taoyuan 333, Taiwan

**Keywords:** cadmium, exposure, dental caries, orthodontic

## Abstract

Background. Previous studies have shown that environmental cadmium exposure could disrupt salivary gland function and is associated with dental caries and reduced bone density. Therefore, this cross-sectional study attempted to determine whether tooth decay with tooth loss following cadmium exposure is associated with some dental or skeletal traits such as malocclusions, sagittal skeletal pattern, and tooth decay. Methods. Between August 2019 and June 2020, 60 orthodontic patients with no history of previous orthodontics, functional appliances, or surgical treatment were examined. The patients were stratified into two groups according to their urine cadmium concentrations: high (>1.06 µg/g creatinine, *n* = 28) or low (<1.06 µg/g creatinine, *n* = 32). Results. The patients were 25.07 ± 4.33 years old, and most were female (female/male: 51/9 or 85%). The skeletal relationship was mainly Class I (48.3%), followed by Class II (35.0%) and Class III (16.7%). Class I molar relationships were found in 46.7% of these patients, Class II molar relationships were found in 15%, and Class III molar relationships were found in 38.3%. The mean decayed, missing, and filled surface (DMFS) score was 8.05 ± 5.54, including 2.03 ± 3.11 for the decayed index, 0.58 ± 1.17 for the missing index, and 5.52 ± 3.92 for the filled index. The mean index of complexity outcome and need (ICON) score was 53.35 ± 9.01. The facial patterns of these patients were within the average low margin (26.65 ± 5.53 for Frankfort–mandibular plane angle (FMA)). There were no significant differences in the above-mentioned dental indices between patients with high urine cadmium concentrations and those with low urine cadmium concentrations. Patients were further stratified into low (<27, *n* = 34), average (27–34, *n* = 23), and high (>34, *n* = 3) FMA groups. There were no statistically significant differences in the urine cadmium concentration among the three groups. Nevertheless, a marginally significant *p*-value of 0.05 for urine cadmium concentration was noted between patients with low FMA and patients with high FMA. Conclusion. This analysis found no association between environmental cadmium exposure and dental indices in our orthodontic patients.

## 1. Introduction

Cadmium is a cumulative toxic non-essential transition metal that poses a serious public health threat [1]. Earlier studies have suggested that cadmium is associated with chronic kidney disease, bone disease, cardiovascular disease, and malignancy [2,3,4,5]. The relationship between cadmium exposure and mortality in populations exposed to high cadmium concentrations, including labors with occupational exposure [6] or residents living in polluted areas, has been confirmed [7]. Among the cadmium pollution incidents in Taiwan, those caused by chemical plants in Taoyuan City from 1983 to 1984 are the most famous. As a result, the nearby farmland was seriously polluted, and it was not replanted. It has been 37 years since the incident, and the half-life of cadmium in the human body has been reported to be approximately 30 years. Most of the patients of Chang Gung Memorial Hospital come from Taoyuan city due to its geographical location. Cadmium measurements conducted on these patients is expected to be able to detect the residual amount of exposure and its effect on local residents.

Cadmium exposure has been reported to inhibit saliva secretion of the parotid gland and reduce the concentration of the main salivary digestive enzyme amylase [8]. It has been suggested that these effects of cadmium may be due to the suppression of acetylcholine release and the destruction of parasympathetic impulses, which play an important role in regulating saliva secretion [8]. The competitive inhibitory effect of cadmium on calcium channels [9,10] and the induction of oxidative stress in the salivary glands can also decrease the function of saliva [11]. Some protein components inside saliva can help reduce the caries rate due to their antibacterial and remineralization functions [12]. Amylase acts on carbohydrates and cleaves starches into smaller polysaccharides, and it has antibacterial properties as well as buffering effects [13]. Proline-rich proteins, statherin and histatins, help with remineralization by increasing the local calcium concentrations due to their affinity for enamel surfaces [14]. Cadmium can inhibit saliva secretion and it can act as a calcium channel blocker that can interfere with calcium metabolism during the formation of tooth enamel. Therefore, cadmium exposure is believed to increase the caries rate [15,16,17].

Cadmium can be consumed through food and obtained from environmental sources such as emissions from mining, smelting, fuel combustion, phosphate fertilizer use, sewage sludge application, disposal of metal wastes, and industrial uses of cadmium in the manufacturing of batteries, pigments, stabilizers, and alloys [17,18,19]. Once cadmium enters the body, it accumulates in the kidneys, liver, and bones and is slowly excreted from the body in urine [19]. A continuous loss of calcium and phosphorus and a disturbed vitamin D metabolism due to renal tubular dysfunction will occur once cadmium enters the body [20]. Cadmium poisoning can affect human organs and cause Itai-itai disease [21]. Bone is one of the target organs for cadmium toxicity [22]. Various changes in the skeletal system characterized by osteopenia, osteoporosis, and/or osteomalacia with an increased incidence of bone fractures are the main unfavorable health effects of chronic environmental exposure to cadmium [23,24,25].

The bone density could vary depending on the amount of contact with cadmium. The facial pattern is believed to be associated with bone density according to the research of facial patterns associated with alveolar bone quality by Li et al. in 2014 and Amini et al. in 2017 [26,27]. There are three basic types of vertical facial patterns: low-angle, average-angle, and high-angle patterns used in orthodontic dentistry [27]. The facial pattern is an important determinant of orthodontic treatment because it can affect treatment goals, prognosis, and plans, and it affects growth prediction, anchoring systems, bite force, and function [26]. Bone density has been reported to be the lowest in patients with high angles among these three patterns [27]. The association between the exposure to cadmium and bone density, and its association with the facial pattern, has led to the question of whether there is a connection between the amount of cadmium exposure and the type of facial pattern.

To evaluate and identify patients with likely success who might take longer to treat, various indices can be used after performing measurements. The indices of complexity outcome and need (ICON) and the decayed, missing, and filled surfaces (DMFS) are used to analyze whether the patient in need has been properly treated at our department. The ICON includes both an esthetic score and an occlusal index of malocclusion severity for the evaluation of treatment need [28]. DMFS is the expression of caries prevalence; thus, it is used to estimate how much dentition is affected by dental caries. A higher score on the DMFS represents more severe dental destruction and therefore might be associated with a higher score on ICON [29].

It has been reported that cadmium exposure results in dental caries, but most of the papers are limited to animal studies and epidemiological data from specific areas. Arora et al. [17] demonstrated that among children with a history of caries in their deciduous teeth, an interquartile range increase in urine cadmium concentration was related to a 17% increase in the number of decayed or filled surfaces. The authors concluded that environmental cadmium exposure could be associated with an increased hazard of dental caries in deciduous teeth. Nevertheless, there was no correlation between urine cadmium concentration and caries in adult teeth [17]. On the other hand, it has also been reported that there is a positive correlation between the cadmium content in the enamel of human premolars and the caries score of permanent teeth. It has also been pointed out that increases in the concentrations of cadmium and tin in human teeth are associated with higher dental caries rates, but the concentrations of copper, lead, and selenium are not [30]. Furthermore, Malara et al. [31] reported that cadmium concentrations in impacted third molars were significantly higher for people living in the relatively polluted Ruda Slaska region than for people living in the Bielsko-Biala region. Combining the above studies, the conclusion is still not clear. The effect of early-life cadmium exposure and repeated lifetime cadmium exposures need to be investigated further.

The influence of cadmium exposure on the dental indices of orthodontic or dental patients is unknown, which initiated our interest in this research. In this study, we hypothesized that cadmium exposure could be relevant to dental caries and bone density. Therefore, this cross-sectional study attempted to determine whether tooth decay with tooth loss following cadmium exposure is associated with some dental or skeletal traits such as malocclusions, sagittal skeletal pattern, and tooth decay.

## 2. Materials and Methods

### 2.1. Ethical Statement

This observational study adhered to the Declaration of Helsinki and was approved by the Medical Ethics Committee of Chang Gung Memorial Hospital, Taiwan. The Institutional Review Board number allocated to the study was 201900197B0. All patients provided written informed consent.

### 2.2. Sample Size Determination

Sample size was determined using G*Power software version 3.1.9.7 (Heinrich-Heine-Universität Düsseldorf, Deutschland, Germany). Based on the study by Karaoğlanoğlu et al. [32], the difference between the DMFS values of subjects whose lactobacillus level in saliva was <105 colony-forming units/mL (*n* = 10) and subjects whose lactobacillus level was ≥105 colony-forming units/mL (*n* = 123) was found to be statistically insignificant (*t* = 1.48, *p* = 0.14). The calculated effect size was 0.4867, and the calculated total sample size was 45 patients under an alpha error of 0.05 and a power of 0.95.

### 2.3. Inclusion and Exclusion Criteria

Between August 2019 and June 2020, all orthodontic patients aged 20 years and older with no history of previous orthodontics, functional appliances, or surgical treatment were eligible for inclusion in this study. Patients with any craniofacial deformities, cleft lip or palate, or previous trauma to the dentofacial structure were excluded from the analysis. Patients were examined for the presence of cadmium in their urine, and their urinary cadmium concentrations were used to assess cadmium exposure [33]. A previous study reported that urinary cadmium is a biomarker of long-term exposure in humans [34].

### 2.4. Assessment of Malocclusion Type, Sagittal Skeletal Pattern, and DMFS

The skeletal relationship between the maxilla and mandible of orthodontic patients was categorized as Class I, Class II, or Class III [35,36,37,38] (Figure 1 and Figure 2). Steiner’s analysis [39] and Tweed analysis [40] were used for the measurements according to the analysis of St. Louis University in the United States, where the first author was trained for her orthodontic graduate program. The dental relationship between the maxilla and mandible of orthodontic patients was defined as molar Class I, Class II, or Class III (Figure 3). The DMFS index was expressed as the total number of teeth that were decayed (D), missing (M), or filled (F) in an individual with permanent dentition. There were five surfaces (facial, lingual, mesial, distal, and occlusal) for the posterior teeth and four surfaces (facial, lingual, mesial, and distal) for the anterior teeth when the DMFS index was calculated.

### 2.5. The Indices of Complexity Outcome and Need (ICON)

ICON is an index of complexity outcome and need that is used for assessing the orthodontic treatment need (Figure 4). ICON has been proposed as a multipurpose occlusal index, including dental components such as crowding measured with the irregularity index/spacing (sagittal dimension), cross bites (transverse dimension), anterior open bite/overbite (vertical dimension), and sagittal posterior occlusion relationship as well as the esthetic component. The esthetic component comprised 10 color photographs showing dentition in the frontal view graded from 1 (most attractive) to 10 (least attractive). The dental indices were documented as numeric values according to the standard. The final ICON score was divided into malocclusion complexity grades (<29 = easy, 29–50 = mild; 51–63 = moderate, 64–77 = difficult, >77 = very difficult). A cutoff point of 43 was set to mark a definite need for orthodontic treatment [41].

### 2.6. Facial Patterns

The Frankfort–mandibular plane angle (FMA) and sella-nasion (SN) to mandibular plane angle are often used to describe these three facial patterns in orthodontics (Figure 5 and Figure 6). FMAs between 27 and 34 are classified as average angles, FMAs less than 27 are classified as low angles, and FMAs larger than 34 are classified as high angles according to the data from the Taiwan Association of Orthodontists. Ricketts E line [42] (a reference line from the chin to the tip of the nose), Holdaway’s H line [43] (a line from the chin to the upper lip), Steiner’s S1 line [42] (a line from the chin to the midpoint bisecting the nasal nostril border line), Burstone’s B line [44] (a line from the chin to the subnasale), Sushner’s S2 line [45] (a line from the soft tissue nasion to soft tissue pogonion), and Merrifield’s profile line and Z angle (an angle formed by a chin-protrusive line intersecting the Frankfort horizontal plane) [37] were used to analyze the facial configurations.

E-lines and Z angles are often used for the evaluation of facial esthetics in our department of Chang Gung Memorial Hospital [37]. The lip protrusion is judged by relating it to the esthetic line (E-line). The E-line (Figure 7) is formed by joining the tip of the nose and soft tissue pogonion, which is the most protrusive point of the bony mandible. The upper lip should be tangent to the line; the lower lip should be tangent or slightly behind the E-line for Caucasians, but there is more variation from 0.2 to 4.3 mm between male and female Taiwanese. The Z angle is another angular measurement for the critical description of the esthetics of the lower face (Figure 8). This angle is formed by the Frankfort plane, and the profile line is formed by a line tangent to the soft tissue chin and to the more prominent lip. The Z angle range is typically 70–80 degrees.

### 2.7. Urine Cadmium Measurement

Urine cadmium measurement by inductively coupled plasma mass spectrometry. Urine specimens were collected and stored in 10 mL metal-free plastic collection tubes. To avoid hydration bias during urine sample collection, urine samples that were overdiluted or overconcentrated (urine creatinine level <10 or >300 mg/dL) were excluded from analysis. Urine specimens were stored at 4 °C. Cadmium was quantified by means of inductively coupled plasma mass spectrometry on a PerkinElmer NexION 350X instrument (MA, USA) and analyzed using a no-gas mode. Urine specimens (500 μL) were diluted (1 + 9) with a 1.5% nitric acid (JT Baker, NJ, USA) solution containing yttrium as an internal standard. The cadmium and yttrium standards were purchased from AccuStandard (CT, USA). The standard range was 1.25 to 40 μg/L. The calibration curve had an R ≥ 0.995. Lypocheck quality controls and urine metal levels 1 and 2 (Bio-Rad Laboratories, Hemel Hempstead, UK) were used and analyzed at the start and end of each analytical run and again after every 10 samples. The lower limit of quantitation (LOQ) for cadmium was 0.3 μg/L. Values below the LOQ were assigned to LOQ for analysis.

### 2.8. Statistical Analysis

Continuous variables were expressed as the mean with a standard deviation, while categorical variables were expressed as numbers and percentages in brackets. All data were tested for normality of distribution and equality of the standard deviation before analysis. Comparisons between groups were performed using Student’s t-test for quantitative variables and Chi-square or Fisher’s exact tests for categorical variables. The criterion for significance was a 95% confidence interval to reject the null hypothesis. All analyses were performed using IBM SPSS Statistics version 20.0(New York, NY, USA).

## 3. Results

Table 1 shows the baseline demographics of the orthodontic patients stratified according to their urine cadmium concentrations as high or low. The patients were 25.07 ± 4.33 years old, and most were female (85%). The average body mass index was 21.73 ± 3.76 kg/m^2^. These 60 patients did not have diabetes mellitus, hypertension, or smoking habits, but a few engaged in alcohol consumption (3.3%) and some had bruxism (31.7%). There were more women in the low urine cadmium group than in the high urine cadmium group (*p* = 0.02). There were no significant differences in any other variables between the groups.

As shown in Table 2, the skeletal relationship of the orthodontic patients was mainly Class I (48.3%), followed by Class II (35.0%) and Class III (16.7%). No significant difference was found between the groups (*p* = 0.74).

Class I molar relationships occurred in 46.7% of these patients, 15% had Class II molar relationships, and 38.3% had Class III molar relationships (Table 3). No significant difference was found between the two groups (*p* = 0.79).

The mean DMFS score was 8.05 ± 5.54, including 2.03 ± 3.11 for the decayed index, 0.58 ± 1.17 for the missing index, and 5.52 ± 3.92 for the filled index (Table 4). There was no significant difference between the patients of the two groups in any dimension of the index (*p* > 0.05).

Among the 60 patients, the average ICON score was 53.35 ± 9.01, which included esthetic, crowding, spacing, crossbite, open bite, overbite, and buccal relationships (Table 5). There was no significant difference between the groups (*p* = 0.59).

As shown in Table 6, the facial patterns of these 60 patients were within the average in the low margin (26.65 ± 5.53 for FMA, 35.62 ± 5.11 for SN-mandibular plane). There were no significant differences between the groups in terms of facial esthetic measurement using the Z angle, E-line to the upper lip, or the E-line to the lower lip (*p* > 0.05).

Table 7 shows the comparison of urine cadmium concentrations in orthodontic patients with different FMAs. Patients were stratified into low (<27, *n* = 34), average (27–34, *n* = 23), and high (>34, *n* = 3) FMA group. There were no statistically significant differences in the urine cadmium concentration among the three groups. Nevertheless, a marginally significant *p* value of 0.05 for urine cadmium concentration was noted between patients with low FMA and patients with high FMA.

## 4. Discussion

The mean age of our subjects who were seeking orthodontic treatment was 25.07 years, range 20–35 years. Younger adults (aged 19–30 years) are more likely to seek orthodontic care according to the data from the Medical Expenditure Panel Survey in the United States [46]. With reference to other national orthodontic patients, their mean age was 26.43 years in the United Kingdom [47]. Trends in adult orthodontic patients have also been reported in Singapore, with a higher proportion of adult patients aged 21–25 years undergoing orthodontic treatment at the National Dental Centre Singapore from 2011 to 2017 [48]. In contrast to the increase in adult patients in other countries, the Australian Society of Orthodontists recommends that children visit a registered specialist orthodontist for an assessment at approximately 7 years of age and that most cases in Australia be treated at ages 12–15. However, there has been an increase in the number of adults seeking orthodontic treatment in most countries. This phenomenon may be attributed to several factors, including esthetics, function and self-esteem, according to the literature findings. In addition to these insights, the improved appearance of braces and the invention of aligners may have increased patients’ motivation [49].

Out of 60 orthodontic patients treated in our department, 51 (85%) of the subjects were female, whereas 15% were male. The majority of orthodontic patients are female in Korea (69.5%) [50], and over 66% of patients in the USA are also female. [46] The motivation of adult female patients seeking orthodontic treatment may include the need for esthetics, function and psychological factors such as self-perception, self-confidence, and avoiding negative thoughts about their teeth and criticism from others [51].

According to our data, there were no significant differences in the DMFS index between patients with high urine cadmium concentrations and those with low urine cadmium concentrations. The average DMFS score was approximately 8.05, which is comparable to that of other countries, such as approximately 6.7 in the United States, 5 in Australia, 7.4 in Iran, 12.0 in Chile, 7.56 in Turkey, 10.8 in Belgium, and 12.5 in Spain [52,53,54,55,56]. Our score is lower than that of European countries, similar to that of Middle-Eastern countries, and it is higher than that of Australia and the United States. Caries expressed as the mean DMFS at our clinic was relatively lower than that in most countries, which might be associated with the nature of the sample. The target patients, consisting of all persons planning for orthodontic treatment, were expected to pay more attention to tooth brushing and to maintain good oral hygiene. In addition, our filling index was higher than that of other countries, revealing that orthodontic patients are more focused on oral health management. The addition of fluoride to drinking water was carried out in the United States and Australia. Their DMFS data suggest that the fluoridation of drinking water can prevent dental caries.

The mean ICON score was 53.35 ± 9.01 for our sample. The ICON score was 44.6 in Iran [57], 63.88 in the Netherlands by calibrated orthodontic [58], 58.9 in South Asia, 57.3 for Americans, 56.6 for Australia, 55.8 for Africa, 52.4 for the Middle East, 50.3 for European countries, and 47.2 for North and East Asia [59]. The cutoff value for treatment needed is 43, according to the paper of N.A. Fox and his coworkers published in the British Dental Journal in 2002 [28]. Our data coincided with this range, and it is believed that the patients who need treatment have been properly treated at Chang Gung Memorial Hospital. The index of ICON offers several advantages, including the ability to assess the complexity of malocclusion and to estimate the treatment need and outcome with relatively high sensitivity and specificity. Despite these advantages, there are still some shortcomings. The cutoff points need to be adjusted in different countries due to the perception of treatment need. The occlusal indices do not cover all occlusal traits, and the assessments are opinion based [60,61]. In terms of facial esthetics other than the data from ICON, we also measured the Z angle and the lip protrusion (E-line to the upper lip and lower lip). The esthetic score from ICON is judged subjectively; in contrast, the Z angle and lip protrusion are obtained objectively by connecting the structures, including the nose, lips, and chin.

There were no significant differences regarding malocclusion in terms of dentition and skeleton between patients with high or low urine cadmium concentrations. There were also no significant differences between the groups in any dimension of the DMFS index. In summary, the previous hypothesis that cadmium may increase the caries rate and therefore may be correlated with subsequent malocclusion cannot be verified. It seems that previous cadmium pollution did not generate residual exposure in the local residents of Taoyuan city, and thereafter, the amount of cadmium detected in our patients was in the lower range of reported human cadmium urine concentrations.

Cadmium pollution can cause bone disease and affect bone quality and density on the basis of several literature searches [62]. In other words, the amount of cadmium can be correlated with bone density. Therefore, the measurement of bone density and quality can be another indicator of exposure to cadmium in Chang Gung Memorial Hospital patients in addition to their urine test. According to previous studies [26,63], bone density is related to the facial shape. Among the three facial types (low, average, and high FMA), the patients with long faces tended to have a decreased bone density. In our research, we performed a statistical analysis for urine cadmium among these three facial types by pairwise comparisons, and indeed, there was a tendency for bone density to decrease among those with longer face shapes (Table 3). There were no statistically significant differences in the urine cadmium concentration among the three FMA groups. Nevertheless, a marginally significant *p* value of 0.05 for urine cadmium concentration was noted between patients with low FMA and patients with high FMA. However, there was still no conclusion about bone quality being associated with facial patterns because the sample sizes in the three groups were not equally distributed (the patients in the group with high FMA were only 3). Essentially, the facial pattern might be influenced by bone thickness and quality. Measuring the amount of cadmium in the body might also provide information on the tendency of facial growth patterns, which can be helpful in selecting orthodontic patients. To verify this statement, the collection of additional samples and computed tomography scans of the bone might be future research directions.

In general, the risk factors for dental caries formation include physical, biological, environmental, behavioral, and lifestyle factors such as cariogenic bacteria, a high sucrose diet, inadequate salivary flow, insufficient fluoride exposure, and poor oral hygiene [64]. Therefore, this study is limited by the small sample size, lack of an equal distribution between sexes, short follow-up duration, and lack of traditional risk factor assessments. The potential limitation to the generalization of these results is the issue of sampling and selection. The sample did not reflect the general population due to the specific geographic scope of the experimental design, which aimed to investigate the cadmium exposure of local residents. A lack of previous research studies on this topic also provides only a limited theoretical foundation for the investigation.

## 5. Conclusions

This analysis found no association between environmental cadmium exposure and dental indices in our orthodontic patients. The investigation in the current study and the previous literature showed a potential toxicity risk of cadmium. In addition, this analysis also showed the necessity for additional in vitro and in vivo studies to comprehensively evaluate the toxicity risk. Finally, in addition to the small sample size, unequal distribution of the sexes, and a short follow-up duration, the study is also limited by a lack of risk factor assessments for dental caries, such as a high sucrose diet, salivary flow, fluoride exposure, and oral hygiene. Further studies are warranted.

## Figures and Tables

**Figure 1 healthcare-09-00413-f001:**
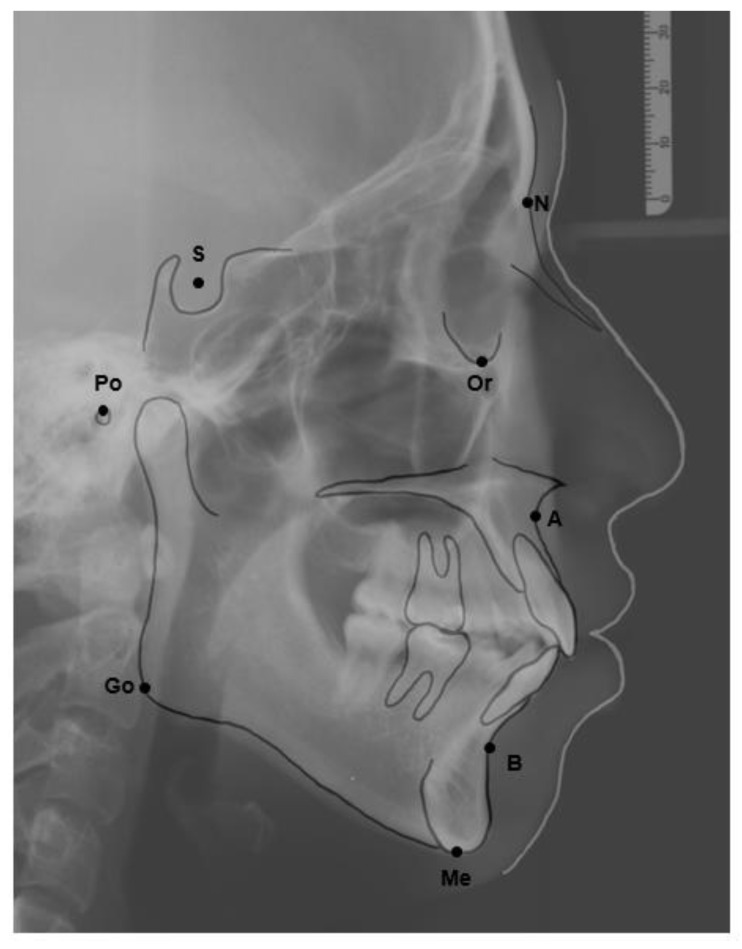
The reference lateral cephalometric radiographic tracing points used in the study. A point: The innermost point on the curvature of the surface of the maxillary bone. B point: The innermost point on the contour of the mandible. Go: gonion: the intersection of the line tangent to the posterior and inferior angle of the mandible. Me: menton, the most inferior point on the mandibular symphysis. N: nasion, the most anterior point of the intersection between the nasal and frontal bones (frontonasal suture). S: sella, the midpoint of the sella turcica. Po: porion, the most superior point of the contour of the external auditory meatus. Or: orbitale, the most inferior point on the margin of the orbit.

**Figure 2 healthcare-09-00413-f002:**
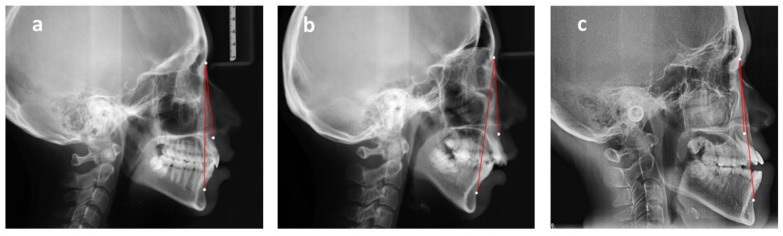
Sagittal skeletal classification based on ANB angle. ANB: The angle is constructed by connecting the A point, nasion, and B point. According to the Steiner analysis, the measurement represents the skeletal relationship between the maxilla and mandible. (**a**) Class I relationship: ANB is between 2 and 4. (**b**) Class II relationship: ANB is greater than 4. (**c**) Class III relationship: ANB is below 2 or negative.

**Figure 3 healthcare-09-00413-f003:**
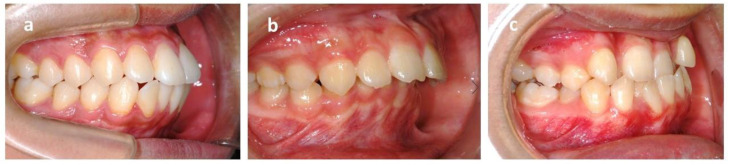
Angle’s classification of malocclusion. (**a**) Class I molar relationship: the mesiobuccal cusp of the upper first molar occludes the mesiobuccal groove of the lower first molar. (**b**) Class II molar relationship: the mesiobuccal cusp of the upper first molar occludes in front of the mesiobuccal groove of the lower first molar. (**c**) Class III molar relationship: the mesiobuccal cusp of the upper first molar occludes behind the mesiobuccal groove of the lower first molar.

**Figure 4 healthcare-09-00413-f004:**
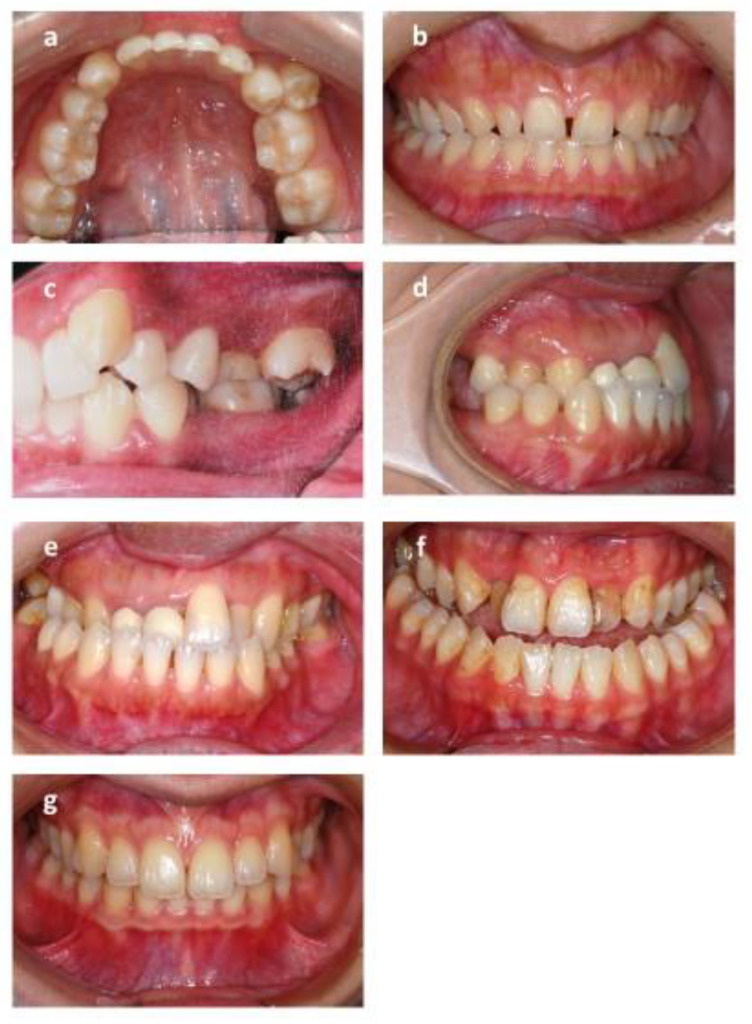
Dental component scale of index of complexity outcome and need (ICON). (**a**) Crowding: crowding occurs when there is a discrepancy between the space required by teeth and the space available in the jawbone. The irregularity index is used for measurement. (**b**) Spacing: the opposite of crowding, the space available is larger than space required. (**c**) Posterior buccal crossbite: buccal cusp of the mandibular dentition occludes lingually to the lingual cusp of the maxillary dentition. (**d**) Posterior lingual crossbite: buccal cusp of the mandibular dentition occludes buccally to the buccal cusp of the maxillary dentition. (**e**) Anterior crossbite: maxillary incisors occlude lingually to mandibular incisors. (**f**) Open bite: there is no vertical overlap between the maxillary and mandibular incisors. (**g**) Overbite: the vertical overlap of the incisors.

**Figure 5 healthcare-09-00413-f005:**
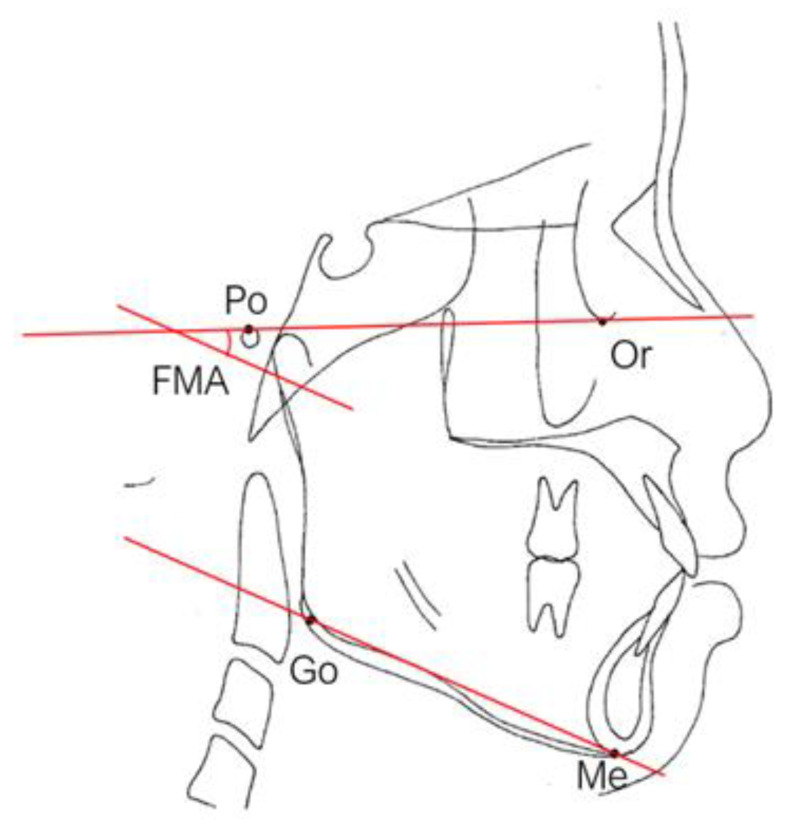
The Frankfort–mandibular plane angle (FMA). The FMA is formed by the intersection of the Frankfort horizontal plane and the mandibular plane.

**Figure 6 healthcare-09-00413-f006:**
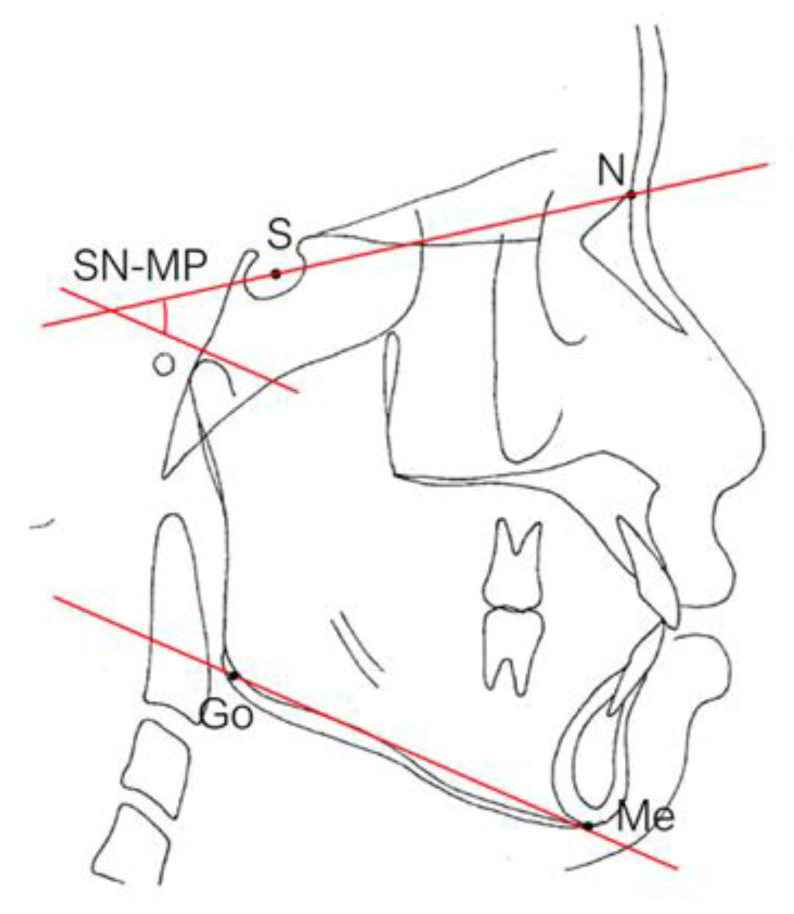
SN plane to the mandibular plane angle (SN–MP). S: sella, the midpoint of the sella turcica. N: nasion, the most anterior point of the intersection between the nasal and frontal bones (frontonasal suture). MP: mandibular plane: formed by connecting Go to Me. Go: gonion: the most posterior and inferior point on the angle of the mandible. Me: Menton, the most inferior point on the chin.

**Figure 7 healthcare-09-00413-f007:**
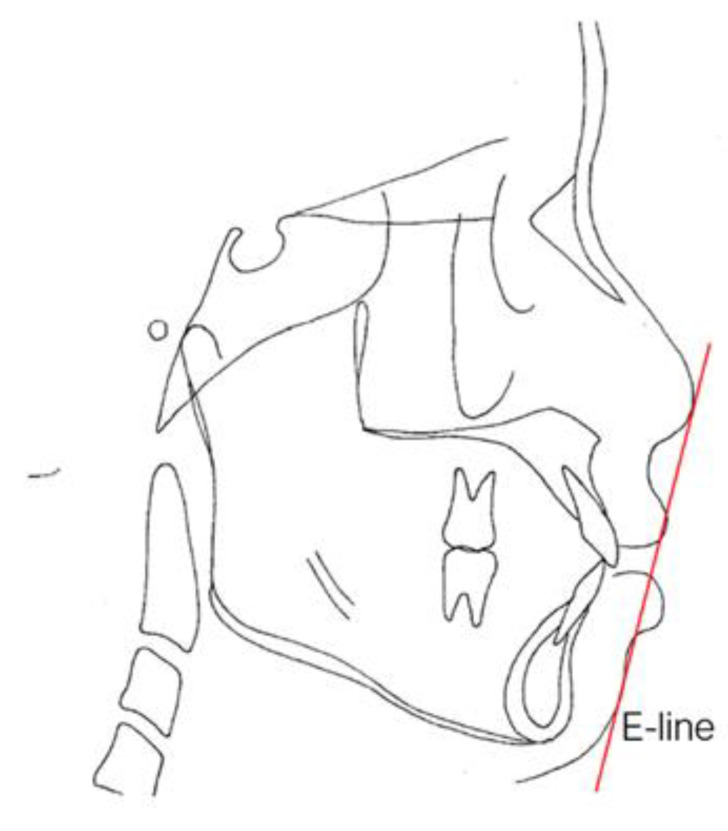
Esthetic line (E-line). The line drawn from the tip of the nose to tip of the chin.

**Figure 8 healthcare-09-00413-f008:**
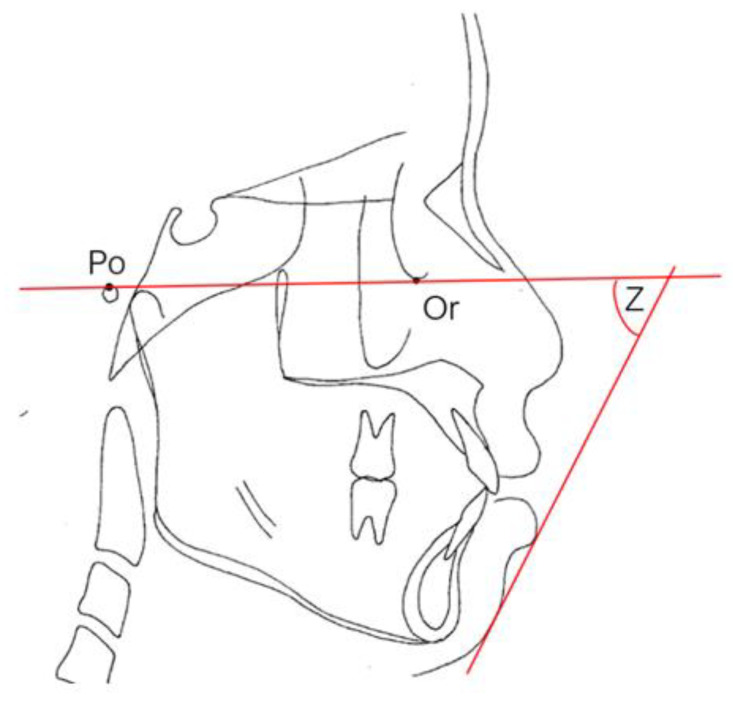
Z angle. The angle formed by the intersection of the Frankfort horizontal plane and a line connecting the soft tissue pogonion and the most protrusive lip.

**Table 1 healthcare-09-00413-t001:** Baseline demographics of orthodontic patients stratified according to their urine cadmium concentrations as high or low (*n* = 60).

Variable	All Patients (*n* = 60)	Patients with High Urine Cadmium Concentration (*n* = 28)	Patients with Low Urine Cadmium Concentration (*n* = 32)	*p* Value
Age, year	25.07 ± 4.33	26.00 ± 5.11	24.25 ± 3.37	0.13
Female, *n* (%)	51 (85.0)	24 (75.0)	27 (96.4)	0.02 *
Body mass index, kg/m^2^	21.73 ± 3.76	21.22 ± 3.68	22.17 ± 3.83	0.34
Diabetes mellitus, *n* (%)	0 (0)	0 (0)	0 (0)	1.00
Hypertension, *n* (%)	0 (0)	0 (0)	0 (0)	1.00
Smoking habit, *n* (%)	0 (0)	0 (0)	0 (0)	1.00
Alcohol consumption, *n* (%)	2 (3.3)	2 (6.3)	0 (0)	0.18
Betel nut chewing, *n* (%)	0 (0)	0 (0)	0 (0)	1.00
Bruxism, *n* (%)	19 (31.7)	13 (40.6)	6 (21.4)	0.11

Note: Patients were stratified into two groups according to their urine cadmium concentrations as high (>1.06 µg/g creatinine) or low (<1.06 µg/g creatinine). Comparisons between groups were performed using Student’s t-test for quantitative variables and Chi-square or Fisher’s exact tests for categorical variables. * *p* < 0.05.

**Table 2 healthcare-09-00413-t002:** Sagittal skeletal relationship of orthodontic patients, stratified according to their urine cadmium concentrations as high or low (*n* = 60).

Variable	All Patients (*n* = 60)	Patients with High Urine Cadmium Concentration (*n* = 28)	Patients with Low Urine Cadmium Concentration (*n* = 32)	*p* Value
Class I, *n* (%)	29 (48.3)	15 (53.6)	14 (43.8)	0.74
Class II, *n* (%)	21 (35.0)	9 (32.1)	12 (37.5)	
Class III, *n* (%)	10 (16.7)	4 (14.3)	6 (18.8)	

Note: Patients were stratified into two groups according to their urine cadmium concentrations as high (>1.06 µg/g creatinine) or low (<1.06 µg/g creatinine).

**Table 3 healthcare-09-00413-t003:** Malocclusion type based on molar relationships of orthodontic patients (Angle’s classification), stratified according to their urine cadmium concentrations as high or low (*n* = 60).

Variable	All Patients (*n* = 60)	Patients with High Urine Cadmium Concentration (*n* = 28)	Patients with Low Urine Cadmium Concentration (*n* = 32)	*p* Value
Class I, *n* (%)	28 (46.7)	12 (42.9)	16 (50.0)	0.79
Class II, *n* (%)	9 (15.0)	4 (14.3)	5 (15.6)	
Class III, *n* (%)	23 (38.3)	12 (42.9)	11 (34.4)	

Note: Patients were stratified into two groups according to their urine cadmium concentrations as high (>1.06 µg/g creatinine) or low (<1.06 µg/g creatinine).

**Table 4 healthcare-09-00413-t004:** Decayed, missing, and filled surface (DMFS) index of orthodontic patients, stratified according to their urine cadmium concentrations as high or low (*n* = 60).

Variable	All Patients (*n* = 60)	Patients with High Urine Cadmium Concentration (*n* = 28)	Patients with Low Urine Cadmium Concentration (*n* = 32)	*p* Value
DMFS index	8.05 ± 5.54	7.71 ± 4.84	8.34 ± 6.15	0.66
Decayed	2.03 ± 3.11	2.18 ± 3.76	1.91 ± 2.45	0.74
Occlusal	0.43 ± 0.81	0.32 ± 0.67	0.55 ± 0.93	0.32
Mesial	1.05 ± 2.05	1.14 ± 2.62	0.97 ± 1.43	0.75
Distal	0.92 ± 1.96	1.11 ± 2.50	0.75 ± 1.34	0.49
Buccal	0.07 ± 0.31	0.11 ± 0.42	0.03 ± 1.78	0.35
Missing	0.58 ± 1.17	0.46 ± 0.92	0.69 ± 1.36	0.47
Filled	5.52 ± 3.92	4.96 ± 2.96	6.00 ± 4.59	0.30
Proximal	0.90 ± 1.50	0.82 ± 1.36	0.97 ± 1.64	0.71
Occlusal	4.58 ± 3.22	4.18 ± 2.68	4.94 ± 3.64	0.37
Mesial	1.10 ± 1.49	1.18 ± 1.44	1.03 ±1.56	0.71
Distal	0.77 ± 1.23	0.75 ± 1.18	0.78 ± 1.29	0.92
Buccal	0.05 ± 0.22	0.04 ± 0.19	0.06 ± 0.25	0.64
Lingual	0.02 ± 0.13	0.00 ± 0.00	0.03 ± 0.18	0.35

Note: Patients were stratified into two groups according to their urine cadmium concentrations as high (>1.06 µg/g creatinine) or low (<1.06 µg/g creatinine). DMFS decayed missing filled teeth.

**Table 5 healthcare-09-00413-t005:** Index of complexity outcome and need (ICON) scores of orthodontic patients, stratified according to their urine cadmium concentrations as high or low (*n* = 60).

Variable	All Patients (*n* = 60)	Patients with High Urine Cadmium Concentration (*n* = 28)	Patients with Low Urine Cadmium Concentration (*n* = 32)	*p* Value
ICON	53.35 ± 9.01	54.04 ± 9.34	52.75 ± 8.82	0.59
Esthetic	5.18 ± 0.43	5.14 ± 0.37	5.22 ± 0.49	0.50
Crowding	0.98 ± 1.05	0.96 ± 1.14	1.00 ± 0.98	0.90
Spacing	0.13 ± 0.47	0.21 ± 0.63	0.06 ± 0.25	0.24
Crossbite	0.55 ± 0.50	0.57 ± 0.50	0.53 ± 0.51	0.76
Open bite	0.13 ± 0.62	0.07 ± 0.38	0.19 ± 0.78	0.48
Overbite	0.93 ± 0.86	1.14 ± 0.93	0.75 ± 0.76	0.08
Buccal	1.35 ± 1.16	1.50 ± 1.14	1.22 ± 1.18	0.35

Note: Patients were stratified into two groups according to their urine cadmium concentrations as high (>1.06 µg/g creatinine) or low (<1.06 µg/g creatinine). ICON index of complexity outcome and need.

**Table 6 healthcare-09-00413-t006:** Selected cephalometric facial measurements in orthodontic patients, stratified according to their urine cadmium concentrations as high or low (*n* = 60).

Variable	All Patients (*n* = 60)	Patients with High Urine Cadmium Concentration (*n* = 28)	Patients with Low Urine Cadmium Concentration (*n* = 32)	*p* Value
FMA	26.65 ± 5.53	26.12 ± 5.68	27.12 ± 5.45	0.49
SN-MP	35.62 ± 5.11	35.02 ± 5.39	36.14 ± 4.88	0.40
Z angle	68.62 ± 9.48	69.30 ± 9.18	68.02 ± 9.85	0.61
E-line U	0.82 ± 3.67	0.71 ± 3.86	0.91 ± 3.55	0.84
E-line L	3.75 ± 4.16	3.11 ± 4.42	4.31 ± 3.90	0.27

Note: Patients were stratified into two groups according to their urine cadmium concentrations as high (>1.06 µg/g creatinine) or low (<1.06 µg/g creatinine). FMA Frankfort–mandibular plane angle; SN-MP Sella-nasion to mandibular plane.

**Table 7 healthcare-09-00413-t007:** Comparison of urine cadmium concentration in orthodontic patients with different FMAs (*n* = 60).

Variable	Patients with Low FMA (*n* = 34)	Patients with Average FMA (*n* = 23)	Patients with High FMA (*n* = 3)
Urine cadmium concentration, µg/g creatinine	1.04 ± 0.60	1.01 ± 0.64	1.74 ± 0.33

Note: FMA: Frankfort–mandibular plane angle. The FMA was described as low < 27, average 27–34, and high >34. The *p* value was 0.05 between patients with low and high PMA; the *p* value was 0.89 between patients with low and average FMA; and the *p* value was 0.06 between patients with average and high FMA.

## Data Availability

The datasets used and analyzed for this study are available from the corresponding author upon request.

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
