# Peer review of "Environmental Cadmium Exposure and Dental Indices in Orthodontic Patients"

_healthcare, 2021, doi:10.3390/healthcare9040413_

Round 1

Reviewer 1 Report

work on grammar and style and presentation, many typos

abstract, add the female/male detail,  selection criteria, random, consecutive?,,,

add briefly why 'urine cadmium' is important

use capital 'C' for Class I,II,II

the literature review is deficient, update

methodology,

any sample size calculation?

add more subheadings for the dental indices, add a few sentences fro each index

discussion, need more information on ICON and its shortcomings as occlusal indices do  not cover all occlusal traits and are opinion based(read, cite, Prog Orthod. 2012 Nov;13(3):314-25. Prog Orthod. 2011 Nov;12(2):132-42. )

you need to discuss that the findings are based on orthodontic patients and not the general public, suggest further studies

the conclusion is very poorly written, revise

some landmark papers are missing, ( mean ICON score in other countries such as Iran 44.6% , add to the paragraph starting line 324, Am J Orthod Dentofacial Orthop2011;140:233-8)

Reviewer 2 Report

This is a study on the relationship between cadmium exposure and oral health indices.

Although it is an interesting topic, many criticisms are nevertheless present:

-In the initial part of the abstract, a general sentence on the risks of cadmium for oral health must be inserted

- The entire numerical list of results must be completely removed from the abstract

-At the end of the introduction section, the null hypotheses of the study must be inserted, which must then be refuted by the results obtained

- The reference to the bioethical committee of the institution to which it belongs is missing. This, which looks like an observational study, necessarily requires this step

- Patient recruitment, both in terms of sample size and inclusion and exclusion criteria, is absolutely lacking. The study design is absolutely not sufficient

-The legends of the trusts 1-4 are too long and should be absolutely reduced

-The results show no statistical significance

-In the conclusions section no consideration is made on how cadmium, as well as other substances, can exert their cytotoxic functions. In this regard, I recommend that you include the following scientific work in the reference section, which could be useful to the reader:

Pagano S, Coniglio M, Valenti C, Negri P, Lombardo G, Costanzi E, Cianetti S, Montaseri A, Marinucci L. Biological effects of resin monomers on oral cell populations: descriptive analysis of literature. Eur J Paediatr Dent. 2019 Sep; 20 (3): 224-232. doi: 10.23804 / ejpd.2019.20.03.11. PMID: 31489823.

-A thorough revision of the English language must be performed

Reviewer 3 Report

The present manuscript aimed to investigate the relationship between environmental cadmium exposure and dental indices to justify cadmium's deleterious effect on adult teeth. To achieve this objective, 60 adult patients were recruited from the orthodontic clinic of Chang Gung Memorial Hospital between August 2019 and June 2020. Patients were examined for the presence of cadmium in their urine and stratified into two groups according to their urine cadmium concentrations, as high (>1.06 ug/g creatinine, n=28) or low (< 1.06 ug/g creatinine, n=32). The authors also determined the skeletal relationship, The decayed, missing, and filled surface (DMFS) score, and the index of complexity outcome and need (ICON) score. None of the scores presented statistical significance between the two groups. Therefore, the authors concluded no association between environmental cadmium exposure and dental indices in our orthodontic patients.

Although simple, the manuscript is well-written. However, I have some questions before I recommend its acceptance

-             The authors must provide the Certified of approval from the Ethical Committee.

-             The authors must perform sample size calculation for their study to demonstrate that the number of patients included is enough for a correct power of the samples

-             The authors should discuss that for caries, many other variables (that were not analyzed in the present manuscript) account for DMFS. The frequency of sucrose ingestion, the quality of oral hygiene performed, and fluoride use have a primordial role in the presence or not of caries. Therefore, the authors must state the limitations of their study.

- Authors should include into their conclusion that many other variables may account for the clinical scores analyzed in the present manuscript and not only the cadmium exposure

Reviewer 4 Report

This article presents a new topic that provides data not known until now by orthodontic science. I appreciate the effort made by the researchers in preparing this article and it follows an appropriate methodology. However, it requires some considerations.
- In the abstract the main objective of the study does not appear, correct it.
- In the bibliographic citations there is a different placement of the punctuation mark before or after the number, correct it, since it always goes after the number. Example: "amylase [1]." and "secretion. [2]”
- In the keywords should go the words that mark the most important line of research. However, cephalometric data that appear to be too technical should be modified. 
- Line 65 quotes:" Therefore, cadmium is believed to increase the caries rate according to most literature. "However, no studies are referenced.
- Lines 89, 90 and 91 are not relevant to the context of the article, nor do they add anything scientific to the paragraph. 
- The introduction makes no reference to why prolonged cadmium exposure may affect orthodontic patients, nor does it cite previous pilot studies that confirm its objective.
- The sample size calculation is not defined, which is relevant in this type of epidemiological study. Exposure to high cadmium content is specific to a region of a country, therefore the sample size should be representative of the general population.
- The inclusion and exclusion criteria for the sample do not appear in the material and method section. Insert according to the described indexes and cephalometric measurements and justify scientifically.
- Figure 1 does not describe the type of cephalometry used for the study. Cephalometric points appear that are not used to classify the sample at skeletal level.
- The described cephalometric points should be referenced bibliographically. Why have you selected these points and not other cephalometrics such as Ricketts, Macnamara or Jarabak?
- You have only taken into account sagittal malocclusions but in the intraoral photographs shown there are patients with transverse problems, have you measured this relationship with cadmium exposure?
- In section 2.3 it talks about facial pattern. The E-line does not and other aesthetic lines are not part of the patient's facial pattern, make a new section or add it.
- In table 1 of results I show variables not previously described in material and method such as diabetes, hypertension, alcohol drinker, even bruxism. Why have you described these variables, what importance do they have for the study in relation to orthodontics.
- In Table 2, the value of the skeletal relationship with cadmium exposure is 0.74, the same value for the three groups. No division was made for each group. The same in table 3.
- What method did you use to measure dental crowding.
- In line 298 there are letters with other fonts different from the text without justifying the wording.
- In the discussion section there is no paragraph that talks about the limitations of the study, which are many (sample size, justification of the work and of the hypotheses, there is no equal distribution between sexes, etc.).
- In the introduction we are told the objective of the study: "Therefore, this study attempted to investigate the relation ship between environmental cadmium exposure and dental indices to justify the deleteri ous effect of cadmium on the adult teeth" However, in the conclusions we are told about the relation of the indices with orthodontics. It is not clear whether the aim of the study is whether tooth decay with tooth loss due to cadmium exposure causes dental or skeletal malocclusions. 

Correct errors in the bibliography.

Round 2

Reviewer 1 Report

abstract

add the age of the patients (mean(SD), male/ female,add the type of study(x-sectional), rewrite 'to determine whether tooth decay with tooth loss following cadmium exposure can cause dental or skeletal 30 malocclusions.' it should be 'to determine whether tooth decay with tooth loss following cadmium exposure is associated with some dental or skeletal traits such as malocclusions, sagittal skeletal pattern, tooth decay.

add that there was a marginally significant P value of 0.05 was noted between patients with low and high FMA.include your findings here. modify the conclusion as well (Patients with high FMA had a higher concentration of Urine cadmium compared to those with low FMA)

introduction, start with a few sentences about cadmium in terms of its chemical characteristics then start with its harmful effect on the human body then the rest

objectives are not clear, this is a crosssections study, you can not mention you are assessing the long term effects, as this need longitudinal study, you just assessed the association with hight cadmium and a few occlusal indices of facial dental features, rewrite your objectives and conclusion

methodology

page 4, correct the following subheadings

section 2.4. DMFS indices, correct to '2.4. assessment of malocclusion type, sagittal skeletal pattern, and DMFT"

sections 2.5 ICON index, correct to '2.5 The indices of complexity  outcome and need (ICON) '

figure 1- correct to' the reference Lateral cephalometric radiographic tracing points used in the study'.

figure 2- correct to' sagittal Skeletal classification based on ANB angle'

figure 3- correct to 'Angle's classification of malocclusion.'

table 2- correct to 'Table 2. sagittal Skeletal relationship of orthodontic patients'

table 3' correct to 'Table 3. malocclusion type based on Molar relationships of orthodontic patients (Angle's classification)

table 5- correct to 'Table 5. ICON scores of orthodontic patients'

table 6- correct to' Table 6. selected cephalometric facial measurements in orthodontic patients'

discussion and conclusion revise, you need to mention that Patients with high FMA had a higher concentration of Urine cadmium compared to those with low FMA, suggest further studies

Reviewer 2 Report

I reccomend work acceptation 

Author Response

Thank you for your help

Reviewer 3 Report

The authrs accepted all of my suggestions

Author Response

Thank you for your help

Reviewer 4 Report

The authors have correctly responded to doubts raised. The article has been improved and can be published. It would be convenient in future research to review these data in the long term. The effort made by the authors is gratefully acknowledged.

Author Response

Thank you for your help